# Do socioeconomic inequities arise during school-based physical activity interventions? An exploratory case study of the GoActive trial

Olivia Alliott ,[1] Hannah Fairbrother,[2] Kirsten Corder,[1] Paul Wilkinson,[3] Esther van Sluijs[1]

[1]MRC Epidemiology Unit, University of Cambridge, Cambridge, UK
[2]Health Sciences School, University of Sheffield, Sheffield, UK
[3]Department of Psychiatry, University of Cambridge, Cambridge, UK

**Correspondence to**
Dr Esther van Sluijs;
esther.vansluijs@mrc-epid.cam.ac.uk

## ABSTRACT

**Objective** To investigate socioeconomic inequities in the intervention and evaluation process of the GoActive school-based physical activity intervention and demonstrate a novel approach to evaluating intervention-related inequalities.

**Design** Exploratory post-hoc secondary data analysis of trial data.

**Setting** The GoActive trial was run in secondary schools across Cambridgeshire and Essex (UK), between September 2016 and July 2018.

**Participants** 13–14 years old adolescents (n=2838, 16 schools).

**Methods** Socioeconomic inequities across six stages in the intervention and evaluation process were evaluated: (1) provision of and access to resources; (2) intervention uptake; (3) intervention effectiveness (accelerometer-assessed moderate-to-vigorous physical activity (MVPA)); (4) long-term compliance; (5) response in evaluation; and (6) impact on health. Data from self-report and objective measures were analysed by individual-level and school-level socioeconomic position (SEP) using a combination of classical hypothesis tests and multilevel regression modelling.

**Results** Stage: (1) There was no difference in the provision of physical activity resources by school-level SEP (eg, quality of facilities (0–3), low=2.6 (0.5); high=2.5 (0.4). (2) Students of low-SEP engaged significantly less with the intervention (eg, website access: low=37.2%; middle=45.4%; high=47.0%; p=0.001). (3) There was a positive intervention effect on MVPA in adolescents of low-SEP (3.13 min/day, 95% CI −1.27 to 7.54, but not middle/high (−1.49; 95% CI −6.54 to 3.57). (4) At 10 months post-intervention, this difference increased (low SEP: 4.90; 95% CI 0.09 to 9.70; middle/high SEP: −2.76; 95% CI −6.78 to 1.26). (5) There was greater non-compliance to evaluation measures among adolescents of low-SEP (eg, % accelerometer compliance (low vs high): baseline: 88.4 vs 92.5; post-intervention: 61.6 vs 69.2; follow-up: 54.5 vs 70.2. (6) The intervention effect on body mass index (BMI) z-score was more favourable in adolescents of low-SEP (low SEP: −0.10; 95% CI −0.19 to 0.00; middle/high: 0.03; 95% CI −0.05 to 0.12).

**Conclusions** These analyses suggest the GoActive intervention had a more favourable positive effect on MVPA and BMI in adolescents of low-SEP, despite lower

## STRENGTHS AND LIMITATIONS OF THIS STUDY

⇒ We present a novel approach to evaluating inequities throughout the intervention process of young people's physical activity interventions.
⇒ While exploratory rather than confirmatory, the small sample size of the low-SEP group raises problems with regard to statistical power.
⇒ The conclusions may be affected by attrition bias, as attrition was largest among participants of a low-SEP.

intervention engagement. However, differential response to evaluation measures may have biassed these conclusions. We demonstrate a novel way of evaluating inequities within young people's physical activity intervention evaluations.

**Trial registration number** ISRCTN31583496.

## INTRODUCTION

The health benefits of physical activity are well-established[1] and physical inactivity has been identified as a major public health concern.[2] Active adolescents experience better present and long-term health and are more likely to become active and healthy adults.[3–5] However, globally over 80% of students aged 11–17 years are insufficiently active to accrue the benefits.[6]

Similar to other health behaviours, disparities in physical activity during adolescence may contribute to inequities in current and future health.[7] Recent review-level evidence highlights the importance of promoting and enabling physical activity among adolescents living in the context of socioeconomic deprivation, who report experiencing more barriers to physical activity when compared with other socioeconomic groups.[8] Despite regularly collecting relevant information at baseline, most controlled trials of physical activity interventions in young people do not

analyse differences in intervention effect across socio-economic groups.[9] This has led to a scarcity of evidence regarding the differential impact of intervention across socioeconomic groups.[9]

Public health literature suggests the extent to which inequities are perpetuated or reduced can depend on the nature of the intervention.[10] 'High-risk strategies' target individuals with a higher risk of developing the disease, whereas population strategies attempt to lower the risk of the entire population by shifting the distribution of underlying risk factors, such as physical inactivity.[11] As a consequence of compulsory education in many countries, the potential for schools to deliver wide-reaching and equitable physical activity interventions has been well documented.[12 13] Taking a population approach, school-based interventions have been studied and deemed successful if average physical activity levels increase.[14] However, population strategies have the potential to inadvertently exacerbate health inequities within a population.[15]

Researchers have begun to consider the potential for interventions to have a differential effect across individuals, commonly named 'intervention generated inequities'.[10] However, across young people's physical activity literature these studies have tended to focus on differential effects by gender.[9] Limited evidence from individual evaluations of physical activity and school-based interventions document socioeconomic inequities negatively impacting those of a low-socioeconomic position (SEP) in the provision of, and access to, interventions and resources,[16 17] intervention uptake,[18] intervention efficacy,[17 19] long-term compliance[20] and differential response in evaluations.[9 21 22]

These previous studies offer examples of various points in the research and intervention process where inequities might emerge. Going forward we propose a broader approach is needed, looking at intervention generated inequities throughout the whole research and intervention process of a single intervention. Based on the work of White *et al*,[10] Love identifies key stages throughout a physical activity intervention where inequities can be introduced.[22] Understanding how inequities might emerge at each of these stages is essential for the development of equitable school-based physical activity interventions, as while inequities at each of these stages could be small, together they may lead to significant inequities in final outcomes.[10]

The aim of this paper is to take a case-study approach to investigate if and how socioeconomic inequities arise during the intervention and evaluation process of the GoActive school-based physical activity intervention. In doing so, we demonstrate a novel way of studying inequities across the intervention implementation and evaluation process that could be applied more broadly.

## METHODS

This paper describes exploratory secondary analyses of the GoActive trial data. These analyses were not detailed in the statistical analysis plan for the main trial analyses, but were guided by a prespecific statistical analysis plan. The GoActive trial was run between September 2016 and July 2018. Ethical approval for the GoActive trial was obtained from the University of Cambridge Psychology Research Ethics Committee (PRE.2015.126). The trial was prospectively registered (ISRCTN31583496).

### Participants and randomisation

Sixteen state-run secondary schools in Cambridgeshire and Essex agreed to participate. All Year 9 students (age 13–14 years) and their parents/carers received written study information. Students provided written assent and parents provided passive informed consent (opt-out consent).[23] School-level randomisation, stratified by the percentage of students eligible for pupil premium funding at each school (below or above county-specific median) and county (Cambridgeshire or Essex), occurred after baseline measurement.[23] Pupil premium funding aims to reduce the effects of deprivation on educational attainment and is used here as a proxy measure for school-level deprivation.[24]

### GoActive intervention

GoActive was a theory-based intervention developed following an evidence-based iterative approach.[23] The primary aim of GoActive was to increase students' moderate-to-vigorous intensity physical activity (MVPA) across the week.[23] GoActive was delivered over 12 weeks to all students in the intervention schools irrespective of whether they participated in study measurements. The control schools followed normal practice.

GoActive was implemented using a tiered-leadership system led by mentors (older students within the school) and supported by peer-elected Year 9 leaders.[23] During the intervention, Year 9 tutor groups chose 2 activities per week from a selection of 20. These activities required little or no equipment and were designed to appeal to a variety of students (including Ultimate Frisbee, Zumba and Hula Hoop). Schools had access to the GoActive intervention website where they could find activity instructions cards which included an overview of each activity, suggested adaptations, safety tips, 'factoids' and a short video.[23] Mentors remained with the class throughout the intervention, whereas peer-leaders changed each week. During the first 6 weeks, additional leadership was provided by a local authority-funded intervention facilitator (health trainers employed by local councils) who continued to provide remote support thereafter.[23]

Teachers were encouraged to dedicate one tutor time per week to do one of the chosen activities as a class. Students could gain points for trying these new activities at any time in or out of school, irrespective of intensity or duration.[23] There was no expectation of time spent in the activities, points were rewarded for taking part. Individual points remained private and students could enter their points at any time on the GoActive website with an individual password and login details. Students

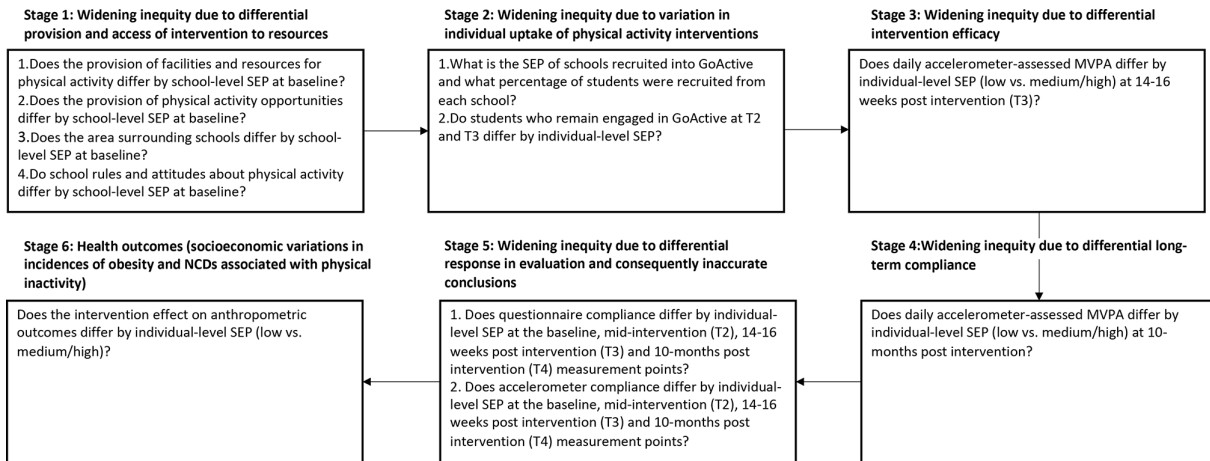

**Figure 1** Intervention stages and accompanying research questions explored throughout the study. Based on the model developed by Love.[22] MVPA, moderate-to-vigorous physical activity; NCDs, non-communicable diseases; SEP, socioeconomic position.

were encouraged to regularly log these points to unlock rewards such as a sports bag, t-shirt or hoodie. While remaining private these points were entered into between-class competitions.[23]

The results of the main GoActive trail analysis reported no overall intervention effect on average daily MVPA.[25] Subgroup analyses conducted as part of the trial evaluation reported a suggestion of a positive intervention effect among students of a low/middle-SEP. Across all MVPA outcomes, those of high-SEP appeared to benefit least when compared with low/middle-SEP students. Full details of the trial methods have been published elsewhere.[23]

### Methodological approach of the current study

As outlined above, we take a case study approach to demonstrate how socioeconomic inequities can be explored throughout the research and intervention process, using the GoActive intervention as an example. As this is an exploratory post-hoc analysis, we operationalised the model proposed by Love to include research questions based on the available GoActive data collected as part of the main GoActive trial (figure 1).[22] For the remainder of this paper, we refer to the stages outlined in figure 1 when describing our research and findings.

This paper focuses on socioeconomic inequities, therefore all of the research questions presented in figure 1 consider SEP. We use individual-level and area-level SEP, as using these different levels are important when evaluating the full contribution of socioeconomic conditions.[26] The relevance of different indicators of SEP is dependent on on the research focus, health outcome and stage in the life course.[26] Taking this approach, we used pupil premium funding (see description in section 2.1) as a school-level indicator of SEP during stages 1 and 2, where the object of analysis is the school, not the individual. Schools were categorised as low-SEP if the percentage of students eligible for pupil premium was below the county-specific mean and high-SEP if the percentage was

above. For the remaining stages, we use an individual-level indicator of SEP derived from the Family Affluence Scale (FAS).[27] In response to the recent review evidence highlighted in the introduction[8] (published after the main GoActive trial) and because of our focus on socio-economic inequities we compare students of low-SEP to students of middle/high-SEP during stages 3, 4 and 5. This is a different approach to that of the main trail which grouped students of low-SEP and middle-SEP together. All measures are described in further detail below.

### Measures

Study measurements were taken at four time points during the GoActive trial: Baseline (BL), mid-intervention (T2; 6 weeks after intervention start), post-intervention (T3; 14–16 weeks after intervention start) and 10-month follow-up (T4; 10 months after intervention end).[25]

A summary of demographic measures and the measures specific to each stage are described below, the best available measures from the trial data were used to address the research questions under each stage. For conciseness, the following shorter titles are applied to each stage: stage 1—provision and access, stage 2—intervention uptake, stage 3—intervention effect, stage 4—long-term compliance, stage 5—evaluation participation and stage 6—health outcomes.

### Demographic measures

Participant descriptive characteristics were self-reported at baseline.[23] Participants reported gender from three response options (male, female and prefer not to say).[25] Individual-level SEP was reported using the FAS, which is composed of six items relating to: (1) family car ownership, (2) holidays, (3) computers, (4) availability of bathrooms, (5) dishwasher ownership and (6) having their own bedroom. These were used as a proxy measure of individual-level socioeconomic position by summing the answers (possible range 0–13), and dividing into predefined affluence groups (low=0–6, middle=7–9 and

high=10–13).[25] Ethnicity was self-reported by participants, who were given 20 response options and an additional free-text option.[25] The reported options were recoded into five categories in accordance with published recommendations: (1) 'white', (2) 'mixed ethnicity' (ie, identifying with multiple ethnicities), (3) 'Asian' (including South Asian and Chinese), (4) 'African and/or Caribbean' and (5) 'other'.[25 28]

### Stage 1: provision and access
Data on the school physical activity policy and social and physical environment were self-reported at baseline by contact teachers (often Physical Education or Year 9 lead) at all schools.[23]

The data were used to highlight the potential for socioeconomic differences in the provision of physical activity opportunities and access to resources at baseline, which may have impacted the delivery of GoActive. These data were collected using a questionnaire previously used in the Year 9 data collection of the Sport Physical Activity and Eating Behaviour, Environmental Determinants in Young People (SPEEDY) study.[29] A list of 16 physical activity facilities available at each school were given a quality rating (0=facility not present, 1=low quality facility, 2=middle quality facility and 3=high quality facility). Ratings were summed and divided by the number of available facilities to give an average quality rating. An average rating was also used to indicate the suitability of the school grounds for sport, informal games and general play across three measures (1=not at all suitable, 2=somewhat suitable and 3=very suitable). The provision of physical activity opportunities was assessed using the extracurricular opportunities on offer at each school derived from a list of 24 (including space to add 'other' activities; one activity=one point, eg, Rounders=1) and weekly hours of PE, measured using an open-ended question where teachers rounded to the nearest half-hour. The suitability of the area around the school for physical activity was assessed on a scale of 1–5 (1=strongly disagree to 5=strongly agree) across three measures, shielding from hedges/trees/fences, maintenance of the grounds and the presence of vandalism. Finally, the school's attitude towards physical activity was assessed using the same 1–5 agreement scale across five measures which included encouraging physical activity at school and outside school, educating about the risks of physical activity and how to practice safe physical activity and encouraging active travel.

Pupil premium was used as a school-level indicator of SEP, which was reported by teachers in the school environment questionnaire.[25]

### Stage 2: intervention uptake
Under stage 2, research questions explore engagement with the GoActive intervention. Recruitment data were used to assess the initial uptake of the intervention by school-level SEP. Evaluation uptake was measured as whether participants provided baseline questionnaire data, which was a requirement for participating in

GoActive.[25] Trained measurement staff checked the questionnaires on completion and helped students complete missing sections.[23] Intervention uptake was assessed using data on students' engagement with the GoActive website as this was the primary method for tracking the activities participants engaged in both in and out of school. This included whether students accessed the GoActive website at any time during the intervention period and was recorded as a categorical variable (accessed vs not). Of the students who did access the website, the number of times they visited and the number of points they logged throughout the intervention were recorded.

### Stage 3: intervention effect
During stage 3, differential intervention efficacy was explored for the GoActive primary outcome, daily accelerometer assessed MVPA at 14–16 weeks post-intervention.[23] Participants were asked to wear a wrist-worn activity monitor (Axivity) assessing acceleration (continuous waveform data) continuously (24 hours a day) for 7 days.[23] The Axivity monitor has been validated to assess energy expenditure and to have increased wear time adherence and acceptability than hip-worn monitors in adolescents.[23 30–32] Monitor output was processed to provide minutes spent in MVPA to be equivalent to ≥2000 ActiGraph counts per minute[23]; further details on accelerometer data processing can be found elsewhere.[25]

### Stage 4: long-term compliance
Stage 4 used accelerometer measurements taken at 10 months post-intervention to reflect long-term compliance to the intervention by exploring compliance to the primary outcome after the intervention period. Average daily minutes of MVPA was used as described above.

### Stage 5: evaluation participation
During stage 5, differential participation in evaluation measures was assessed using compliance with questionnaire and accelerometer measures. Questionnaire compliance was defined as whether participants provided questionnaire data at each measurement occasion. Research staff working on the GoActive study recorded whether a questionnaire for each participant had been completed and checked at each measurement point. Accelerometer compliance was determined as whether participants provided valid accelerometer data at each measurement point. In line with the main GoActive trial analysis, participants were required to provide 6 hours of wear time from a possible 42 hours in each daytime quadrant: morning (06:00 to 12:00), afternoon (12:00 to 18:00), evening (20:00 to 24:00) and night (24:00 to 06:00).[25]

### Stage 6: health outcomes
Related health outcomes were explored during stage 6 using anthropometric measures. During a school site visit, trained measurement staff conducted the following measures according to standardised operation procedures: height (m), weight (kg), waist circumference (cm)

and bioimpedance to assess body fat percentage (%).[23] Body mass index (BMI) SD scores were calculated from height and weight data (i.e. weight/height$^2$ (kg/m$^2$)) and categorised according to age and gender standardised International Obesity Task Force thresholds.[25]

## Analysis

Characteristics of the sample were described using mean, SD and frequency values. Data from all measurement points were included across the analyses described below and were stratified by either individual-level or school-level SEP. All included analyses were exploratory, but guided by an analysis plan developed prior to release of the data.

Research questions under stages 1 (provision and access) and 2 (intervention uptake) used self-reported data from the school environment and student questionnaires. Data were explored using simple tabulations, graphical techniques and basic summary statistics and analysed using Kruskal-Wallis one-way analysis of variance by school-level SEP. This test was selected due to the skewness of the data. P values were adjusted for ties as the same values occurred in more than one sample. For further analyses under stage 2, website access by the intervention group was also explored using Pearson's $\chi^2$. Of those who accessed the website, differences in the number of visits and points logged by individual-level SEP were analysed using the Kruskal-Wallis test as described above.

Research questions under stages 3 (intervention effect) and 4 (long-term compliance) were explored using accelerometer assessed MVPA, interaction analyses were run to examine if the effect of the independent variable (intervention vs control) on the dependent variable (daily average MVPA) differed by individual-level SEP, following statistical procedures from the main GoActive analyses.[23] For MVPA at T3 and T4 (ie, the primary outcome), the intervention effect, representing the baseline-adjusted difference in change from baseline between the intervention and control groups, was estimated from a linear regression model, including randomisation group, baseline value of the outcome (i.e. analysis of covariance), the randomisation stratifiers (ie, pupil premium funding and county) and an interaction between individual-level SEP and group allocation. Models were also run separately for low and middle/high socioeconomic groups to assess intervention effects within subgroups. Robust SEs were calculated to allow for the non-independence of individuals within schools.

Under stage 5 (evaluation participation), we examine differential response to evaluation measures by individual-level SEP. We examined accelerometer compliance and self-report compliance (eg, questionnaire completion vs no completion) using Pearson's $\chi^2$.

Stage 6 (health outcomes) was explored using anthropometric outcomes. Interaction analyses were used to examine if the effect of the independent variable (intervention vs control) on the dependent variable (BMI, waist circumference or body fat) differed by individual-level SEP, separate analyses were run for each anthropometric variable following the same analytical approach as stages 3 and 4.

All analyses were conducted using Stata V.15.1 software.

## Patient and public involvement

None for the purpose of this secondary data analysis.

## RESULTS

### Sample description

A total of 2838 students provided baseline questionnaire data. Table 1 provides an overview of baseline characteristics by individual-level SEP. Overall, mean age was 13.3 (SD 0.4) years, just over half of the participants were men (51.4%) and the majority of the participants were of white British ethnicity (84.7%). Fewer participants were of a low-SEP (14.0%), than of middle-SEP (42.5%) and high-SEP (43.5%).

### Main analyses

#### Stage 1: provision and access

Table 2 shows that regardless of school-level SEP, teachers reported their schools to be suitable for physical activity at baseline. Differences between the provision of and access to physical activity facilities by school-level SEP were tested, but none were identified as statistically significant with p values >0.05.

#### Stage 2: intervention uptake

Table 2 provides a breakdown of recruitment by school-level SEP, suggesting that a lower proportion of students from low-SEP were recruited into the GoActive trial, particularly in high-SEP schools.

Table 3 presents the uptake of the GoActive intervention by individual-level SEP using website engagement. The results show that significantly fewer students of low-SEP than middle-SEP and high-SEP accessed the GoActive intervention website. There was no difference in engagement found for those who did access the website.

#### Stage 3: intervention effect

Table 4 shows the moderating effect of SEP on the effectiveness of the GoActive intervention on average daily minutes of MVPA. The results of the interaction analysis suggest at the post-intervention measurement the intervention effect in students of middle/high-SEP was 4.56 (95% CI −9.56 to 0.41) min/day less MVPA than students of low-SEP. However, subgroup analyses did not show statistically significant effects in either group.

#### Stage 4: long-term compliance

At 10 months post intervention, the difference in intervention effect increased to −7.53 (95% CI −12.89 to −2.17) min/day MVPA in favour of participants of low-SEP (table 4). Subsequent stratified analyses showed a positive intervention effect in participants of a low-SEP (4.90; 95% CI 0.09 to 9.70) but not those of middle/high-SEP.

**Table 1** Baseline descriptive characteristics by individual-level SEP

|  | Low-SEP | Middle-SEP | High-SEP |
|---|---|---|---|
| **N (%)** | | | |
| Participant | 398 (14.0) | 1206 (42.5) | 1234 (43.5) |
| **Gender** | | | |
| Male | 196 (6.9) | 598 (21.1) | 684 (24.1) |
| Female | 202 (7.1) | 608 (21.4) | 550 (19.4) |
| **Ethnic group** | | | |
| White | 319 (11.3) | 1032 (36.5) | 1071 (37.8) |
| Mixed/multiple ethnic background | 32 (1.1) | 73 (2.6) | 76 (2.7) |
| Asian or Asian British | 20 (0.7) | 52 (1.8) | 36 (1.3) |
| Black or black British | 16 (0.6) | 30 (1.1) | 24 (0.8) |
| Other ethnic group | 10 (0.4) | 16 (0.6) | 22 (0.7) |
| **Mean (SD)** | | | |
| Age | 13.2 (0.4) | 13.3 (0.4) | 13.3 (0.5) |
| BMI | 21.1 (4.3) | 20.5 (3.7) | 20.1 (3.5) |
| Body fat % | 22.1 (10.2) | 21.3 (10.1) | 19.9 (9.7) |
| Waist circumference | 71.7 (11.1) | 70.5 (9.7) | 69.0 (8.9) |

BMI, body mass index; SEP, socioeconomic position.

*Stage 5: evaluation participation*

Figure 2A shows that questionnaire compliance decreased throughout the intervention across all socioeconomic groups. The figure also shows an association between individual-level SEP and questionnaire compliance (lower compliance among students of low-SEP). Differences in compliance increased with time from T2 to T3 to T4 (T2: $X^2$ = 23.45, p=0.00; T3: $X^2$ = 15.25, p=0.00; T4: $X^2$ = 43.88, p=0.00). Figure 2B shows this trend was also observed for accelerometer compliance (BL: $X^2$ = 8.90, p=0.02; T3: $X^2$ = 8.12, p=0.02; T4: $X^2$ = 33.65, p=0.00).

*Stage 6: health outcomes*

Table 4 shows an indication (p=0.09) of a more favourable intervention effect on the BMI z-score in participants of low-SEP, however the interaction term was not statistically significant. No interaction effects were observed for waist circumference or body fat. Subsequent stratified analyses suggest a favourable intervention effect on BMI z-score among adolescents of low-SEP when compared with the control condition (low SEP: –0.10; 95% CI –0.19 to 0.00), but not for those of middle/high-SEP (middle/high: 0.03; 95% CI –0.05 to 0.12).

See online supplemental table 1 for mean physical activity and anthropometric outcomes by SEP and randomisation group at each measurement point.

## DISCUSSION

Taking a case-study approach we investigated if and how socioeconomic inequities arose during the intervention and evaluation process of a school-based physical activity intervention called GoActive. In doing so, we present a novel approach to analysing young people's physical activity interventions from an equality lens. The findings described below demonstrate the benefit of taking this approach to intervention evaluation, providing insight beyond the main trial analysis.

We discuss our main findings in relation to three key elements: intervention context and engagement (stages 1 and 2), intervention effectiveness (stages 3, 4 and 6) and intervention recruitment and evaluation (stages 2 and 5).

### Intervention context and engagement

Our finding that school-level SEP did not appear to influence the school physical activity environment at baseline, contrasts with previous research highlighting socioeconomic inequities in school physical activity provision and resources.[8 33 34] While it is likely this could be the result of the small sample size of included schools (n=16) and resultant limited power to show significant differences, this could also be due to the UK context of GoActive, where national and local policy, such as the Schools Premises Regulations, impose minimum standards for school sports grounds and facilities.[35 36] In addition to extra funding available for low-SEP schools, such a pupil premium funding[24] which could be spent on the provision of physical activity resources and opportunities.

In relation to engagement, significantly fewer adolescents of low-SEP accessed the GoActive website. Of those who did, a graded effect was observed with adolescents of low-SEP engaging the least. One explanation for this, as highlighted in previous research, is that students living in the context of socioeconomic deprivation interact differently with the school environment (eg, the use of

**Table 2** Physical activity environment and recruitment by school-level SEP

| | Schools of low-SEP (N=8) | Schools of high-SEP (N=8) |
|---|---|---|
| | Mean (SD) | Mean (SD) |
| **Physical activity environment** | | |
| **School level measure (possible range)** | | |
| Quality of school physical activity facilities (0–3) | 2.6 (0.5) | 2.5 (0.4) |
| Suitability of school grounds for physical activity (3–9) | 8.3 (1.5) | 8.0 (1.4) |
| Extra-curricular opportunities for physical activity (0–25) | 11.0 (2.2) | 12.5 (3.7) |
| Weekly hours of physical education (0+) | 2.0 (0.0) | 2.2 (.4) |
| Area around school suitable for physical activity (3–15) | 11.9 (2.2) | 12.8 (1.2) |
| School attitude towards physical activity (5–25) | 18 (6.0) | 19.3 (6.3) |
| **Recruitment rates** | | |
| Number of Year 9 students at baseline (N) | 1648 | 1759 |
| Recruited at baseline N (%) | 1369 (83.1) | 1469 (83.5) |
| **Students from each family affluence group by school-level SEP** | **N (%)** | **N (%)** |
| Low individual-SEP | 266 (19.4) | 132 (9.0) |
| Middle individual-SEP | 598 (43.7) | 608 (41.4) |
| High individual-SEP | 505 (36.9) | 729 (49.6) |

Higher scores=more favourable facilities, opportunities, environment or attitude.
SEP, socioeconomic position.

equipment, fostering of autonomy, competence and relatedness, update of extracurricular opportunities) potentially impacting their engagement with GoActive.[37] [38]

Furthermore, review evidence reports that adolescents of low-SEP experience multiple barriers to engaging in physical activity interventions, including digital exclusion.[14]

## Intervention effectiveness

Despite apparently lower engagement, our exploratory analyses suggest that participants of a low-SEP responded more favourably to GoActive, with a difference in effect of 7.53 min/day at 10 months post-intervention to participants of a middle/high-SEP. It may be that students of a low-SEP had a lower engagement with the website but were more engaged with other elements of the intervention that we have no data on. The observed intervention effect of ~5 min of MVPA per day may be important for health,[39] and was the targeted effect in the main GoActive trial. A similar pattern of effect was also observed for BMI z-score.[23] Overall, these findings support the potential for school-based interventions to reduce inequities in physical activity and obesity. It is possible more deprived students particularly benefitted from the chance to try the variety of new activities offered during the GoActive intervention.[8] [25] This is especially promising given the stark socioeconomically patterned inequities in overweight and obesity in the UK and other high-income countries.[8] [40]

Our choice to treat adolescents of low-SEP as an independent group was based on recent review evidence that their experiences of physical activity notably differ to those of middle-SEP and high-SEP, highlighting the value of looking at them as a separate group.[8] By doing so, our findings add to the main trial moderation analyses where participants of low-SEP and middle-SEP were grouped, suggesting the intervention effect was primarily experienced among low-SEP adolescents. While the approach initially taken was prespecified and common among existing literature,[23] mainly due to the small sample size of low-SEP groups, these exploratory analyses suggest that important differences in effect may be overlooked when taking this approach.

## Intervention recruitment and evaluation

Recruitment data showed that 14% of those participating in the GoActive trial were of a low-SEP and the majority of these students attended low-SEP schools. In the East of England, data from the Family Resources Survey (2016–2019) reports 19.5% of young people were living in poverty at the time GoActive was delivered.[41] It is possible that this is due to the small sample of 16 schools that are

**Table 3** Website access and engagement of students in the intervention group

| | Low-SEP N=235 | Middle-SEP N=670 | High-SEP N=606 | $X^2$ | df | P value (adjusted for ties) |
|---|---|---|---|---|---|---|
| Accessed the website N (%) | 94 (40.0) | 304 (45.4) | 315 (52.0) | 16.52 | 2 | 0.00 |
| Mean website points (SD) | 49.8 (123.1) | 53.2 (85.1) | 55.0 (87.8) | 0.53 | 2 | 0.77 |
| Mean website visits (SD) | 14.2 (28.1) | 15.5 (21.0) | 16.0 (22.8) | 0.74 | 2 | 0.69 |

SEP, socioeconomic position.

**Table 4** Intervention effect on daily accelerometer assessed MVPA at 14–16 weeks and 10 months post intervention and on anthropometric measures at 10 months post intervention

| | B | 95% CI | P value | Model N |
|---|---|---|---|---|
| **14–16 weeks post intervention** | | | | |
| **MVPA** | | | | |
| *Interaction term* | | | | |
| Intervention×SEP | −4.56 | −9.56 to 0.41 | 0.069 | 1878 |
| *Stratified analysis* | | | | |
| Low-SEP | 3.13 | −1.27 to 7.54 | 0.150 | 241 |
| Middle/high-SEP | −1.49 | −6.54 to 3.57 | 0.540 | 1637 |
| **10 months post intervention** | | | | |
| **MVPA** | | | | |
| *Interaction term* | | | | |
| Intervention×SEP | −7.53 | −12.89 to −2.17 | 0.009 | 1785 |
| *Stratified analysis* | | | | |
| Low-SEP | 4.90 | 0.09 to 9.70 | 0.046 | 203 |
| Middle/high-SEP | −2.76 | −6.78 to 1.26 | 0.164 | 1582 |
| **BMI z-score** | | | | |
| *Interaction effect* | | | | |
| Intervention×SEP | 0.12 | −0.02 to 0.26 | 0.096 | 2070 |
| *Stratified analysis* | | | | |
| Low-SEP | −0.10 | −0.19 to 0.0 | 0.055 | 247 |
| Middle/high-SEP | 0.03 | −0.05 to 0.12 | 0.413 | 1823 |
| **Body fat (%)** | | | | |
| *Interaction term* | | | | |
| Intervention×SEP | 1.09 | −0.63 to 2.81 | 0.198 | 1873 |
| *Stratified analysis* | | | | |
| Low-SEP | −0.69 | −3.17 to 1.78 | 0.560 | 216 |
| Middle/high-SEP | 0.41 | −0.75 to 1.57 | 0.464 | 1657 |
| **Waist circumference (cm)** | | | | |
| *Interaction term* | | | | |
| Intervention×SEP | 0.73 | −0.68 to 2.15 | 0.287 | 2089 |
| *Stratified analysis* | | | | |
| Low-SEP | −0.71 | −1.64 to 1.30 | 0.808 | 249 |
| Middle/high-SEP | 0.56 | −0.17 to 1.30 | 0.124 | 1840 |

Note: All models adjusted for school-level SEP, county and school clustering; MVPA models also adjusted for baseline MVPA.
BMI, body mass index ; MVPA, moderate-to-vigorous physical activity; SEP, socioeconomic position.

unlikely to be representative of the county. Furthermore, while 'living in poverty' is a different indicator of SEP than family affluence, measures of SEP are shown to be highly correlated.[26 42] It is therefore worth considering, when comparing these percentages, that adolescents of low-SEP might be under-represented in the overall GoActive sample, aligning with evidence that socioeconomically disadvantaged groups are 'hard to reach' and recruit into research.[43] Of those recruited into GoActive, inequities in study evaluation measures were observed. These results are consistent with previously reported socioeconomic patterns in response to survey evaluation measures.[10 44 45] Higher accelerometer non-response has also been reported among socioeconomically-deprived children,[46 47] however, there is a lack of research looking at socioeconomic patterning in accelerometer compliance among adolescent populations.

Based on these findings, it is important to acknowledge that our analyses were conducted using a small subset of students of low-SEP which may result in bias in our conclusions. It is possible that differential engagement and response to evaluation measures resulted in a subset of students of low-SEP who were not reflective of the group more broadly, impacting the generalisability of our

A

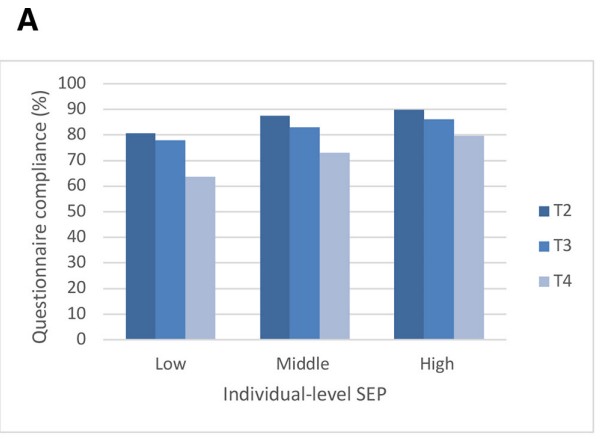

B

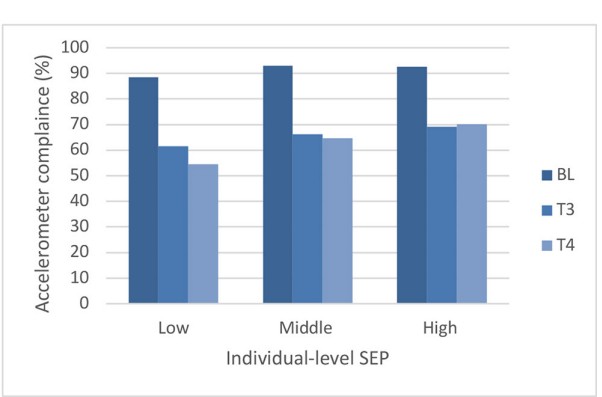

*SEP=Socioeconomic position*

**Figure 2** Compliance to study evaluation measures throughout the GoActive trial by individual level-SEP, indicated by (A) percentage of students proving questionnaire data; and (B) percentage of students providing accelerometer data. BL, baseline.

results. Furthermore, it is possible this may have impacted the results of our analyses under stages 3 and 4, where those who remained in GoActive are more likely to be those who got most out of it.

### Strengths and limitations

Previous research has begun to look at differential effectiveness using the primary outcome of the intervention.[9] To our knowledge, this is the first paper to provide an example of how inequities can be explored throughout the intervention and research process of young people's physical activity interventions. Taking a stage-based approach we highlight differential engagement in specific components of the GoActive intervention, including accessing the GoActive website and in response to evaluation measures. We further highlight the potential of school-based interventions to reduce inequities in MVPA and obesity. Further strengths include the diversity of data collected during the GoActive trial which allowed us to build a more holistic picture of inequities during the trial and the use of device-measured MVPA, which aligns with public health research recommendations for the objective and comprehensive evaluation of health promotion programmes.[25 48]

While presenting our result as exploratory, rather than confirmatory, we acknowledge the small sample size of the low-SEP group and school-level data raises problems with regards to statistical power.[49] The subjective quantification of school environment features may have given rise to self-report biases.[37] It is possible that teachers' reported acceptability of physical activity resources was relative to school-level deprivation, with teachers at high-SEP schools expecting a higher standard of resources and facilities. It is also suggested that some activity types (eg, biking, stair walking) and intensities can be misclassified by wrist-worn accelerometers.[50] If these behaviours are also socioeconomically patterned, this may have led to an underestimation or overestimation of the difference in effect between subgroups. It is also possible that differential access to

computers outside of school hours may have impacted engagement with the GoActive website.[14] Further limitations include the relative lack of participants of a low-SEP and of non-white ethnicity.[25]

It is acknowledged that this is an exploratory post-hoc study of the main trial data collected for GoActive. It is therefore presented as an example of one approach to exploring intervention-generated inequities throughout the intervention and evaluation process. With this in mind, the operationalisation of each stage (figure 1) and the resultant analyses were based on the best available data from the GoActive trial and not what would ideally be the most appropriate data to address each stage of Love's model. Data were not available to address other relevant questions, such as whether schools has access to facilities specifically needed to run GoActive (rather than general facilities) (to address stage 1), the SEP of schools that agreed to participate versus those who did not (to address stage 2), the role the intervention development process could have played in the uptake of and engagement with the intervention (to address stage 5) or the cumulative effects of inequities across the stages of Love's model. A further limitation was not being able to use the focus group and interview data collected as part of the GoActive process evaluation, as information was not available on the SEP of the participants involved. To properly address each stage of Love's model, the stages need to be considered and embedded in the research design.

### Recommendation for future research and practice

As highlighted above, this paper presents a case-study example of how to analyse young people's physical activity interventions with an equity lens. Drawing on the stages developed by Love, a framework for future studies to apply, adapt and develop is provided.[22] While the paper focuses on young people's physical activity interventions, the application of this approach more broadly is encouraged. The data required for such a comprehensive analysis should be considered during the design stage of

future interventions and trials. This will help prevent the development and implementation of unequitable interventions, making better use of public resources.[51] The financial and resource requirement for running sufficiently large trials to detect a main intervention effect and differential effects between subgroups are acknowledged.[9] To tackle this, Love *et al* have previously recommended encouraging coordinated efforts towards fewer, high-quality, large trials, adequately powered to address questions of differential effectiveness.[9] This study echoes this statement, continuing to solely add evidence on overall effectiveness will continue to limit the evidence-base and our understanding from progressing.

Mobilising the approach presented in this project for existing intervention strategies will further help develop our understanding of why current interventions appear to be ineffective in tackling physical inactivity during adolescence.[12] In addition to developing our understanding of the most appropriate data to address each stage of the model, going forward it would be useful to apply this approach to a range of trials to provide researchers and public health professionals with further examples of how to assess inequities at each stage, generating ideas within the research community and continuing to develop this approach.

The results of this stage-based analysis show the potential for universal school-based physical activity interventions to positively impact socioeconomically deprived students (who remained participating in the trial), reducing inequities. Importantly this contradicts the common assumption that interventions generate or exacerbate inequities.[9] The results also demonstrate how intervention components that require individual agency, such as accessing the GoActive website, can exacerbate inequities.[52 53] This should be considered in the development and implementation of school policy, especially in schools with a high proportion of students of a low-SEP. Due to the exploratory nature of this study, it would be beneficial for future research to further study the potential benefit of school-based physical activity interventions for students of low-SEP. It may be useful to explore the application of easily accessible interventions, such as the Daily Mile, to a secondary school setting.[54]

Recruiting and retaining participants of a low-SEP can be challenging, which means they are often underrepresented in research.[14] To increase the reach of interventions and to be able to conduct statistically powered subgroup analyses, the development of active and targeted recruitment of adolescents living in the context of socioeconomic deprivation is an important area for future research. The lack of representation of low-SEP groups in intervention development is an opportunity for growth within school-based interventions and an important area to be considered in the development and evaluation of interventions. Strategies are also needed to better engage these adolescents in the research process, for example, involving them in the design and research process through patient and public involvement.[55]

## CONCLUSION

This was an exploratory study exploring whether and how socioeconomic inequities might arise throughout a school-based physical activity intervention. We demonstrate how the GoActive trial positively affected the physical activity and BMI of low-SEP students. However, differential engagement in the intervention and response to evaluation measures may have biassed these conclusions. The continued development and evaluation of school-based interventions from an equity lens is essential as we move out of the COVID-19 pandemic, where disparities in school-based physical activity were exacerbated.[56]

**Contributors** OA conceived the study in collaboration with EvS and HF, OA is the guarantor for the study. KC led the GoActive trial and provide feedback on this secondary analysis of the GoActive data, as did PW.

**Funding** OA is funded by the National Institute for Health and Care Research (NIHR) School for Public Health Research (SPHR), Grant Reference Number PD-SPH-2015. HF is funded through the NIHR School for Public Health Children, young people and families programme. The work of EvS is supported by the Medical Research Council (grant number MC_UU_00006/5). The GoActive trial was funded by the NIHR Public Health Research Programme (https://www.nihr.ac.uk/explore-nihr/funding-programmes/public-healthresearch.htm; award number: 13/90/18) and undertaken under the auspices of the Centre for Diet and Activity Research (CEDAR), a UKCRC Public Health Research Centre of Excellence. The GoActive trial was also supported by NIHR Biomedical Research Centre Cambridge: Nutrition, Diet and Lifestyle Research Theme (Grant IS-BRC-1215-20014) to KC and EvS. The funders had no role in the GoActive study design, data collection and analysis, decision to publish or preparation of the manuscript. The views expressed in this article are those of the author(s) and the funders had no role in the design of the study, data analysis, data interpretation or in writing the manuscript.

**Competing interests** None declared.

**Patient and public involvement** Patients and/or the public were not involved in the design, or conduct, or reporting, or dissemination plans of this research.

**Patient consent for publication** Not applicable.

**Ethics approval** Not applicable.

**Provenance and peer review** Not commissioned; externally peer reviewed.

**Data availability statement** All data relevant to the study are included in the article or uploaded as supplementary information. The authors do not have the authority to share the data that support the findings of this study and the data are not openly available because of ethical and legal considerations. Non-identifiable data can be made available to bona fide researchers on submission of a reasonable request to datasharing@mrc-epid.cam.ac.uk. The principles and processes for accessing and sharing data are outlined in the MRC Epidemiology Unit Data Access and Data Sharing Policy.

**ORCID iD**
Olivia Alliott http://orcid.org/0000-0002-1835-6852

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
