## [Reviewer comments · BMJ Open]

ARTICLE DETAILS

TITLE (PROVISIONAL)	Do socioeconomic inequities arise during school-based physical activity interventions? An exploratory case study of the GoActive trial
AUTHORS	Alliott, Olivia; Fairbrother, Hannah; Corder, Kirsten; Wilkinson, Paul; van Sluijs, Esther

VERSION 1 – REVIEW

REVIEWER	Harrington, Deirdre University of Strathclyde, Psychological sciences and health
REVIEW RETURNED	12-Sep-2022

GENERAL COMMENTS	This paper is a secondary analysis of data collected in a large school-based physical activity (PA) trial. The paper is operationalising a previous published model on points in the intervention process where inequalities may be generated. The authors take a case study approach and use what data they have available to fit with the six stages in the model. How inequalities in school-based PA trials is a crucial concept for interventionists, evaluators, funders and stakeholders to be aware of. This paper (or at least the concept behind it) would benefit from a number of major additions to move the field forward. From the outset I was unsure about whether the authors are proposing an identification of inequalities or a method to ensure known and potential inequalities are tackled at each stage of the research lifecycle. Terminology used adds to the confusion. On page 6 line 27 the paper talks about inequalities being “introduced” while line 33 says “emerged”. Page 23 line 37 then says “analyse.” Section 2.3 sets the methods up as a way to “assess” inequalities rather than look at how they are introduced and by whom or how they can be addressed and by whom. I believe the authors are proposing that there are different ways that inequalities can occur in school-based PA trials. My interpretation is that some inequalities may be driven by decisions during intervention development (e.g. not tailoring the intervention to certain underserved groups). Some are due to the intervention deployment (e.g. burden on already pushed school staff to provide support) or in the evaluation (e.g. doing evaluations without taking into account headwear worn for religious or cultural observations, translations of materials for parents who may need assistance with written English). As an example using Figure 1. Which of these are introduced by the research team and which are uncovered and need to be identified and addressed before and intervention is developed and
---

delivered? I guess I am asking how many are due to researchers or how many happen anyway and researchers need to be aware?

****Major comments****

Page 10 Line 40 mentions a questionnaire to “assess intervention resources” but it seems like the questionnaire is actually looking at the school facilities and environment for PA in general rather than those specifically needed for GoActive to run. Stage 1 is an assessment of facilities that are relevant and crucial for intervention success. For GoActive that could be sports hall, staff time/workload/experience, which are most relevant?

Page 10 Line 45 says “Ratings were summed and divided by the number of available facilities to give an average quality rating.” Is that a standard way to calculate using that questionnaire? Would there be another way to do it if you wanted to assess the available facilities and provision suited to the intervention (rather than just the quality of facilities in general). Just be clear on what the purpose of this questionnaire is and whether this questionnaire has limitations in assessing the provision relevant to the GoActive trial.

Related to the last point. How are factors in the questionnaire such as extracurricular sport and PE hours relevant to the delivery of GoActive?

Can you mention inequalities related to staff? This could be due to staffing levels to support the GoActive intervention, staff workload, skills of staff.

Stage 2 – intervention uptake. Could there also be inequalities due to the schools that take on the intervention. That is how I would define uptake. You could look at uptake in terms of what was the SES of the schools that agreed to participate versus those that didn't. I would hypothesize that higher SEP schools got involved as there was more staff available to support it running and support the time for evaluations. Are you actually measuring engagement with the intervention resources rather than uptake in Stage 2?

Mention the pupil-facing website in the methods when the intervention is described as only the website for the school is talked about. As access/visits/clicks are a main measure in the case study then it is good to know what that website actually involves and how integral it was to the intervention. From the methods described it would seem that the activities within school are more central to the intervention, would that be fair?

Are there cumulative effects of stages as each of the stages are not independent? The intervention effect could be unequal due to factors in Stage 3 that then will persist into Stages 4 onwards. So once these exist at an early stage they cannot be undone in subsequent stages or they would be expected to still persist in later stages?

I know the authors are not proposing that this is the ideal way to assess inequalities at every stage. There is no indication from the authors of whether each of the ways of measuring for each step is optimal or recommended. What might be more useful is to choose a range of trials to do the same as what has been done for

	GoActive so readers then have a number of examples of how to assess inequalities at each stage. Therefore, it would be useful to have some other examples of how this has been done or could be done ideally to generate ideas within the research community. Also, having an indication of limitations to the GoActive measures proposed at each step. Opening of the discussion says “in doing so, we present a novel approach to analysing young people’s physical activity intervention from an equality lens.” I would argue that measures available post hoc may not be ideal as it was admitted that some data were not available or not collected. This case study is an example of how it has been done and I suggest detailing the limitations to the measures that were available. “We provide a framework for future studies to apply, adapt and develop” but I would like a comment on what would or could be an ideal measure as each stage. Page 20 line 14 – “Our findings demonstrate the benefit of taking this approach to intervention evaluation” and Page 22 Line 57 “We demonstrate how taking this approach provides additional insight for the development of effective and equitable interventions.” What are these benefits? Yes, it is a secondary analysis of data which is good and does allow you to critique the intervention methods and approach a bit more. What is this insight and how can it be applied? What is the main impact that your findings would have on GoActive it were run again? Page 20 line 34 Page 20 – “highlighting socioeconomic disparities in school physical activity provision and resources.” In what direction? Often low SEP schools will have better facilities due to additional investments in these areas of the UK. Please comment on the direction within all of the studies referenced. Page 20 line 44 – “being exposed to similar school environments, significantly fewer adolescents of low-SEP accessed the GoActive website” What is the hypothesised link between adequate school facilities for PA and ability or desire of young people to access a PA website. Is there not any other measure of engagement by the participants e.g. attendance at sessions? “Inequity” is used in the figure but “inequalities” used throughout the paper. These are two different concepts and should not be used interchangeably.
--	---

REVIEWER	Johnson, Ashleigh San Diego State University, Exercise and Nutritional Sciences
REVIEW RETURNED	23-Sep-2022

GENERAL COMMENTS	Overall Thoughts: Thank you for the opportunity to review this interesting work. This manuscript describes findings from an exploratory study examining the potential socioeconomic differences in the intervention process for the GoActive school-based physical activity intervention. The findings showed that there were differences in engagement and effect of the intervention by socioeconomic groups. The manuscript is well-written and examines an important topic. However, additional clarity and justification around the methodological approach is needed to support interpretation of the findings. These comments and additional feedback are discussed in more detail below.
--

	Major Comments: Methods: Be sure that all analyses are described in the analytic plan, while clarifying which analyses were a priori versus post-hoc. There are some analyses that are not discussed in the methods but are mentioned in the results. More information is needed regarding intervention components and engagement. The peer- and mentor-led physical activities appear to be the primary component of the intervention, and the website provided additional resources for participants. Is there information available about the group activities (e.g., level of engagement from students, activities conducted, characteristics of mentors/peers) that may influence participants' interest in the intervention? Also, it is stated that schools had access to the intervention website – does this mean that students could only access the website at school? Or were they encouraged/able to access the information at home? If at home, what percentage of youth had access to computers at home? Consider providing more rationale for selecting website access as the measure of engagement. The measure for long-term compliance (MVPA at 10 months post intervention) examines more the intervention effect than compliance to the intervention. A better measure of long-term compliance would be long-term intervention uptake (e.g., use of the website). Please provide clarification regarding selection of this measure for long-term compliance. Results: How were schools and individuals categorized as low-, middle-, and high SEP? How many schools were categorized as low versus high SEP? What were the student characteristics of these schools overall (versus just the intervention participants)? This will help provide context for whether the participants were representative of the schools they were attending. Discussion: One area that is not fully addressed is intervention development, and the role this could play in intervention uptake and subsequent stages. It is unclear based from the Corder et al. paper whether students of lower SES were included in the GoActive intervention development process. This has major implications for whether the intervention would address stated interests/needs of students across the socioeconomic groups. Lack of representation of low SES groups in intervention development is an opportunity for growth within school-based PA interventions and contributes to the dearth of literature the authors state in the introduction. Although this is briefly addressed, I recommend the authors expand on this concept in the discussion/conclusion section. Minor Comments: Be sure to define all acronyms throughout the manuscript. Throughout the manuscript, be sure there is consistency with the stated objective. For example, in the Abstract, the main objective of the paper is first stated as examining the “implementation and evaluation” of the GoActive intervention. However, the abstract methods then state that the intervention and research process were examined, and this seems to be more in line with what was done. The authors may consider calling this the “intervention process”, versus stating “implementation” for example. Implementation science is a process unto itself, and may be misleading to readers, particularly because there was actually little data presented about the implementation process (e.g.,
--	---

	characteristics of the implementers, how well the intervention components, etc. were implemented). Introduction: Page 5, Line 11: The phrase “life chances” is unclear. Perhaps “all-cause mortality”? Page 5, Lines 31-34: I agree that there is a benefit to understanding the differential effects of PA interventions on youth from different socioeconomic groups. I think THIS is more the gap in the literature, rather than whether they would benefit from PA interventions (which is pretty well established). Page 6, Lines 7-9: Please clarify whether this refers to school-based PA interventions specifically. Page 6, Lines 9-16: Please elaborate on these socioeconomic inequalities – specifically, which socioeconomic groups it negatively affects. It can be assumed that those of low SES are negatively affected, but it would be helpful to clearly state this as it is the main premise of the study. Methods: Please include the dates of the study. Be sure to clarify which analyses were a priori versus post hoc. For example, on Page 8, Lines 32-34: were the subgroup group analyses post hoc? Page 9, Lines 14-18: Please clarify whether this review of the literature and decision making was post hoc. Additionally, please expand on the “recent review evidence” and how that informed the comparison by low-SEP versus middle/high-SEP. Page 10, Line 36: Please state in what population/setting this questionnaire has been previously used, as well as any other psychometric properties that are available. Was there an option for respondents to state, “I don’t know”? Page 10, Line 34: Please briefly describe the characteristics of contact teachers (e.g., role, years taught, age, etc.). Results: Page 15, Lines 56-60: Examining differences by school-level SEP should be included in the analysis plan. Page 19, Line 34: Change “man” to “mean” Discussion: Page 23, Line 23: Change “wrist-worm” to “wrist-worn” Page 23, Line 27: Is “different” supposed to be “difference”? Figure 1: The first and third box appear to state the same thing. Please update if needed. Also, please be sure to define “NCD” within the figure legend. Tables and Figures: Please define SEP, SD, MVPA, and BMI Tables: Please take care to clarify whether “Low-SEP, Middle SEP, and High SEP” refers to individuals or schools, particularly with Table 3. Supplemental Table 1: please define acronyms and include n’s References: Consider updating the references to include more recent work, unless those included are seminal works.
--	---

VERSION 1 – AUTHOR RESPONSE

Reviewer 1			
6.	Page 10 Line 40 mentions a questionnaire to “assess intervention resources” but it seems like the questionnaire is actually looking at the school facilities and environment for PA in general rather than those specifically needed for GoActive to run. Stage 1 is an assessment of facilities that are relevant and crucial for intervention success. For GoActive that could be sports hall, staff time/workload/experience, which are most relevant?	We agree with the point that the questionnaire was not developed specifically to assess access to intervention resources at baseline. In response, we have rephrased our description of this as follows: “Data on the school physical activity policy and social and physical environment were self-reported at baseline by contact teachers (often Physical Education or Year 9 lead) at all schools.²³ The data were used to highlight the potential for socioeconomic differences in the provision of physical activity opportunities and access to resources at baseline, which may have impacted the delivery of GoActive. These data were collected using a questionnaire previously used in the Year 9 data collection of the Sport Physical Activity and Eating Behaviour, Environmental Determinants in Young People (SPEEDY) study.²⁹” We also agree that there are many other measures such as ‘staff time/workload/experience’ which could be more relevant. Due to the post hoc exploratory nature of this study we did not have access to this data. This has been addressed in the Strengths and Limitations section (4.4) of the Discussion.	333-341
7.	Page 10 Line 45 says “Ratings were summed and divided by the number of available facilities to give an average quality rating.” Is that a standard way to calculate using that questionnaire? Would there be another way to do it if you wanted to assess the available facilities and provision suited to the intervention (rather than just the quality of facilities in general). Just be clear	The decision to calculate the quality of facilities in this way was a pragmatic decision based on the information provided in the questionnaires and the data available to us. We acknowledge this as a limitation in the general discussion as follows: “With this in mind, the operationalisation of each stage (Figure 2) and the resultant analyses were based on the best available data from the GoActive trial and not what would ideally be the most appropriate data to address each stage of Love’s model (Figure 1).” However, GoActive was a very flexible intervention aiming to encourage students to try out and engage in a wide variety of activities. We therefore decided that a broad measure of the	717-720

	on what the purpose of this questionnaire is and whether this questionnaire has limitations in assessing the provision relevant to the GoActive trial.	physical activity environment would be most suited as the facilities and resources that would enable effective participation in GoActive were varied.	
8.	Related to the last point. How are factors in the questionnaire such as extracurricular sport and PE hours relevant to the delivery of GoActive?	These factors were included as an indicator of physical activity opportunities offered by schools at baseline as these have the potential to impact the schools delivery of the GoActive intervention. As highlighted in Figure 2, one of the proposed research questions under Stage 1 was “Does the provision of physical activity opportunities differ by school-level SEP at baseline?”, we included extracurricular sport and PE hours as measures of existing opportunities and as indicators of the schools’ provision and access to physical activity opportunities at baseline. This has been clarified in the methods section as follows: “The provision of physical activity opportunities was assessed using the extracurricular opportunities on offer at each school derived from a list of 24 (including space to add ‘other’ activities; one activity = one point e.g. Rounders = 1) and weekly hours of PE, measured using an open-ended question where teachers rounded to the nearest half-hour.” For clarity the following description has been added to section 2.4.3: “The data were used to highlight the potential for socioeconomic differences in the provision of physical activity opportunities and access to resources at baseline, which may have impacted the delivery of GoActive. These data were collected using a questionnaire previously used in the Year 9 data collection of the Sport Physical Activity and Eating Behaviour, Environmental Determinants in Young People (SPEEDY) study.²⁹”	347-351 337-341
9.	Can you mention inequalities related to staff? This could be due to staffing levels to support the GoActive	Whilst we agree this is an important point, unfortunately these data were not collected as part of the GoActive trial. We refer to our Strengths and Limitations section (4.4) where we have added a more explicit explanation of the limitations of the measures used in this study.	717-731

	intervention, staff workload, skills of staff.		
10	Stage 2 – intervention uptake. Could there also be inequalities due to the schools that take on the intervention. That is how I would define uptake. You could look at uptake in terms of what was the SES of the schools that agreed to participate versus those that didn't. I would hypothesize that higher SEP schools got involved as there was more staff available to support it running and support the time for evaluations. Are you actually measuring engagement with the intervention resources rather than uptake in Stage 2?	We agree with this point and that information on schools who take on the intervention is valuable information regarding inequalities in intervention uptake. As highlighted during our response to previous comments, unfortunately we do not have data on schools who did not take in the intervention. We hope that our acknowledgement of the limitations of the available data (highlighted above) and discussion of this in section 4.4 addresses this comment.	717-731
11	Mention the pupil-facing website in the methods when the intervention is described as only the website for the school is talked about. As access/visits/clicks are a main measure in the case study then it is good to know what that website actually involves and how integral it was to the intervention. From the methods described it would seem that the activities within school are more central to the intervention, would that be fair?	Please see lines 252-254 (highlighted in the text) which describe how students were encouraged to log activity points on the GoActive website. To clarify, there was no school/teacher facing website. Information of website use has been added and highlighted in the manuscript. The website is now explained as follows: “Teachers were encouraged to dedicate one tutor time per week to do one of the chosen activities as a class. Students could gain points for trying these new activities at any time in or out of school, irrespective of intensity or duration.²³ There was no expectation of time spent in the activities, points were rewarded for taking part. Individual points remained private and students could enter their points at any time on the GoActive website with an individual password and login details. Students were encouraged to regularly log these points to unlock rewards such as a sports bag, t-shirt, or hoodie. Whilst remaining private these points were entered into between-class competitions.²³”	247-254

		Whilst we endeavour to provide as much information as possible to understand the analyses undertaken in this paper, we have also tried to keep the paper succinct to enhance readability. We encourage interested readers to read related papers to gain a full understanding of the GoActive intervention and its evaluation.	
12	Are there cumulative effects of stages as each of the stages are not independent? The intervention effect could be unequal due to factors in Stage 3 that then will persist into Stages 4 onwards. So once these exist at an early stage they cannot be undone in subsequent stages or they would be expected to still persist in later stages?	We agree with the potential for cumulative effects across intervention and evaluation stages. We explore cumulative impacts, to some extent, across the different stages by looking at different measurement points e.g. questionnaire and accelerometer compliance at T2, T3 and T4 and MVPA and anthropometric measures at 14-16 weeks and 10-month post intervention. We also account for the observed attrition bias in drawing our conclusions around effectiveness. However, we acknowledge that data were not available to analyse the cumulative effects of each stage across the model or the impact of one stage on the next. As with previous comments, we hope we have addressed this comment by highlighting it in the study limitations as follows: “Data were not available to address other relevant questions, such as whether schools has access to facilities specifically needed to run GoActive (rather than general facilities) (to address stage 1), the SEP of schools that agreed to participate vs those who did not (to address Stage 2), the role the intervention development process could have played in the uptake of and engagement with the intervention (to address stage 5) or the cumulative effects of inequities across the stages of Love’s model.”	722-728
13	I know the authors are not proposing that this is the ideal way to assess inequalities at every stage. There is no indication from the authors of whether each of the ways of measuring for each step is optimal or recommended. What might be more useful is	This comment has been addressed in strengths and limitations section and recommendations for future research. We agree that the ways for measuring a operationalising each step taken in this paper might not be optimal and highlight this. As follows: “It is acknowledged that this is an exploratory post-hoc study of the main trial data collected for GoActive. It is therefore presented as an example of one approach to exploring intervention generated inequities throughout	717-722

	to choose a range of trials to do the same as what has been done for GoActive so readers then have a number of examples of how to assess inequalities at each stage. Therefore, it would be useful to have some other examples of how this has been done or could be done ideally to generate ideas within the research community. Also, having an indication of limitations to the GoActive measures proposed at each step.	the intervention and evaluation process. With this in mind, the operationalisation of each stage (Figure 2) and the resultant analyses were based on the best available data from the GoActive trial and not what would ideally be the most appropriate data to address each stage of Love’s model (Figure 1). We agree it would be useful to have a range of examples for how this has or could be ideally done. We address this in the recommendation for future research as follows: “Mobilising the approach presented in this project for existing intervention strategies will further help develop our understanding of why current interventions appear to be ineffective in tackling physical inactivity during adolescence.¹⁰ In addition to developing our understanding of the most appropriate data to address each stage of the model. Going forward it might be useful to apply this approach to a range of trials to provide researchers and public health professionals with further examples of how to assess inequalities at each stage, generating ideas within the research community and continuing to develop this approach.”	752-759
14	Opening of the discussion says “in doing so, we present a novel approach to analysing young people’s physical activity intervention from an equality lens.” I would argue that measures available post hoc may not be ideal as it was admitted that some data were not available or not collected. This case study is an example of how it has been done and I suggest detailing the limitations to the measures that were available. “We provide a framework for future studies to apply, adapt and develop” but I would like a comment on what	As highlighted above, we thank the reviewer for this comment and have now addressed this in the limitation section as follows: “It is acknowledged that this is an exploratory post-hoc study of the main trial data collected for GoActive. It is therefore presented as an example of one approach to exploring intervention generated inequities throughout the intervention and evaluation process. With this in mind, the operationalisation of each stage (Figure 2) and the resultant analyses were based on the best available data from the GoActive trial and not what would ideally be the most appropriate data to address each stage of Love’s model (Figure 1). Data were not available to address other relevant questions, such as whether schools has access to facilities specifically needed to run GoActive (rather than general facilities) (to address stage 1), the SEP of schools that agreed to participate vs those who did not (to address Stage 2), the role the intervention development process could have played in the	717-731

	would or could be an ideal measure as each stage.	uptake of and engagement with the intervention (to address stage 5) or the cumulative effects of inequities across the stages of Love’s model. A further limitation was not being able to use the focus group and interview data collected as part of the GoActive process evaluation, as information was not available on the SEP of the participants involved. To properly address each stage of Love’s model, the stages need to be considered and embedded in the research design.” Due to the large variation in intervention and evaluation designs we consider it not possible nor desirable to indicate what measures should be included as this will be context-specific. Instead, we encourage researchers to plan ahead for similar analyses and consider the measures and data required to address the questions that are pertinent to their trial and context.	
15	Page 20 line 14 – “Our findings demonstrate the benefit of taking this approach to intervention evaluation” and Page 22 Line 57 “We demonstrate how taking this approach provides additional insight for the development of effective and equitable interventions.” What are these benefits? Yes, it is a secondary analysis of data which is good and does allow you to critique the intervention methods and approach a bit more. What is this insight and how can it be applied? What is the main impact that your findings would have on GoActive if it were run again?	We thank the reviewer for this comment and would like to highlight what we view our analyses add to the main trial analyses. Firstly, we highlight that participants of a low-SEP responded more favourable to GoActive in terms of MVPA. These findings add to the main trial moderation analysis where participants of low-and middle-SEP were grouped, our results suggest the intervention effect was primarily experienced among low-SEP adolescents. Our analyses further suggest that important differences in effect may be overlooked when combining middle- and low-SEP groups and highlight the need for sufficiently powered trials to detect differential intervention effects, as highlighted in our recommendation for future research. Highlighted in the text as follows: “Our choice to treat adolescents of low-SEP as an independent group was based on recent review evidence that their experiences of physical activity notably differ to those of middle- and high-SEP, highlighting the value of looking at them as a separate group.⁸ By doing so, our findings add to the main trial moderation analyses where participants of low-and middle-SEP were grouped, suggesting the intervention effect was primarily experienced among low-SEP adolescents. While the approach initially taken was pre-specified and common among existing literature,²³ mainly due to the small	643-651

		sample size of low-SEP groups, these exploratory analyses suggest that important differences in effect may be overlooked when taking this approach.” Furthermore, by taking a stage based approach we highlight differential engagement in specific components of the intervention including accessing the GoActive website and in response to evaluation measures including questionnaire response and accelerometer compliance. These findings highlight intervention components which have the potential to exacerbate inequalities which should be explored in the future development of equitable physical activity interventions. Highlighted in the text as follows: “Previous research has begun to look at differential effectiveness using the primary outcome of the intervention.⁹ To our knowledge, this is the first paper to provide an example of how inequalities can be explored throughout the intervention and research process of young people’s physical activity interventions. Taking a stage-based approach we highlight differential engagement in specific components of the GoActive intervention, including accessing the GoActive website and in response to evaluation measures. We further highlight the potential of school-based interventions to reduce inequities in MVPA and obesity.”	685-691
16	Page 20 line 34 Page 20 – “highlighting socioeconomic disparities in school physical activity provision and resources.” In what direction? Often low SEP schools will have better facilities due to additional investments in these areas of the UK. Please comment on the direction within all of the studies referenced.	“socioeconomic disparities” has been rephrased to “socioeconomic inequities” to highlight the direction of the relationship, as demonstrate below. “Our finding that school-level SEP did not appear to influence the school physical activity environment at baseline, contrasts with previous research highlighting socioeconomic inequities in school physical activity provision and resources.^{6,31,32}” It is also acknowledged that low-SEP schools in the UK have access to additional funding/support as highlighted in the comment. “In addition to extra funding available for low-SEP schools, such a pupil premium funding²⁴ which	602-604 608-610

		could be spent on the provision of physical activity resources and opportunities.”	
17	Page 20 line 44 – “being exposed to similar school environments, significantly fewer adolescents of low-SEP accessed the GoActive website” What is the hypothesised link between adequate school facilities for PA and ability or desire of young people to access a PA website. Is there not any other measure of engagement by the participants e.g. attendance at sessions?	This sentence has been rephrased as follows, “In relation to engagement, significantly fewer adolescents of low-SEP accessed the GoActive website.” Due to the design and delivery of the GoActive intervention (targeting a whole year group with delivery at form-group level), there was no other measure of GoActive engagement available. Similar to the approach taken to define a per-protocol subgroup for our main trial analyses, we considered access to the intervention website a key indicator of engagement as this gave access to rewards.	612-613
18	“Inequity” is used in the figure but “inequalities” used throughout the paper. These are two different concepts and should not be used interchangeably.	We thank the reviewer for this comment and have now consistently used “inequities” throughout the paper to reflect unfair differences which could arise during school-based physical activity interventions, which if understood could be avoided.	
Reviewer 2			
19	Be sure that all analyses are described in the analytic plan, while clarifying which analyses were a priori versus post-hoc. There are some analyses that are not discussed in the methods but are mentioned in the results.	All analyses included in this paper are exploratory and were not part of the original Statistical Analysis Plan of the GoActive trial. However, a detailed analysis plan was drawn up prior to data release and this was adhered to in the analyses. Detailed as follows: “All included analyses were exploratory, but guided by an analysis plan developed prior to release of the data.” We were unable to identify analyses not described in the analysis section, and would be grateful if the reviewer could indicate specifically which results they have identified that were not appropriately described in the methods.	432-433
20	More information is needed regarding intervention components and engagement. The	We have expanded our description of the intervention. As highlighted in response to comment 10, we have had to balance the need for detailed information with manuscript length and	

	peer- and mentor-led physical activities appear to be the primary component of the intervention, and the website provided additional resources for participants. Is there information available about the group activities (e.g., level of engagement from students, activities conducted, characteristics of mentors/peers) that may influence participants' interest in the intervention?	therefore encourage interested readers to access related papers for more detail about the GoActive intervention and its main evaluation. We acknowledge that additional/other measures of intervention engagement would have been beneficial to our analyses. Unfortunately information on engagement in group activities which could be analysed by SEP was not available. As part of the main trial analysis qualitative interviews were conducted as part of the process evaluation which speak to this. However, unfortunately information is not available on the SEP of participants who provided qualitative data. We have addressed this in the limitations section by highlighting how we were unable to use this data, as follows: “A further limitation was not being able to use the focus group and interview data collected as part of the GoActive process evaluation, as information was not available on the SEP of the participants involved. To properly address each stage of Love’s model, the stages need to be considered and embedded in the research design.” As highlighted in our response to the comments from reviewer one, we have added to the limitations section a statement about the limitations of the measures used in this study to address each stage of Love’s model.	728-731
21	Also, it is stated that schools had access to the intervention website – does this mean that students could only access the website at school? Or were they encouraged/able to access the information at home? If at home, what percentage of youth had access to computers at home? Consider providing more rationale for selecting website access as the measure of engagement.	Additional information has been added to the methods section on students access to the GoActive website as follows: “Teachers were encouraged to dedicate one tutor time per week to do one of the chosen activities as a class. Students could gain points for trying these new activities at any time in or out of school, irrespective of intensity or duration.²³ There was no expectation of time spent in the activities, points were rewarded for taking part. Individual points remained private and students could enter their points at any time on the GoActive website with an individual password and login details. Students were encouraged to regularly log these points to unlock rewards such as a sports bag, t-shirt, or hoodie. Whilst remaining private these points were entered into between-class competitions.²³”	247-254 369-371

		An additional justification has been added for selection website access as the measure of engagement: “Intervention uptake was assessed using data on students’ engagement with the GoActive website as this was the primary method for tracking the activities participants engaged in both in and out of school.” The decision to use a website to allow participants to track points and claim rewards was based on feedback received during the development and early testing of the GoActive intervention. Whilst we acknowledge that there may be differential access to computers at home, all schools had computers available that the students could access to engage with GoActive.	
22	The measure for long-term compliance (MVPA at 10 months post intervention) examines more the intervention effect than compliance to the intervention. A better measure of long-term compliance would be long-term intervention uptake (e.g., use of the website). Please provide clarification regarding selection of this measure for long-term compliance.	We thank the reviewer for their comment on this and once again acknowledge that the measures used are not necessarily the most desirable measures to address the stages of Love’s model. MVPA at 10-months post intervention was chosen as a measure of long-term compliance to reflect long-term compliance to the intervention by exploring compliance to the primary outcome after the intervention period. A clearer justification for this measure has been added to the methods section as follows: “Stage 4 used accelerometer measurements taken at 10-months post-intervention to reflect long-term compliance to the intervention by exploring compliance to the primary outcome after the intervention period. Average daily minutes of MVPA was used as described above.”	396-398
23	How were schools and individuals categorized as low-, middle-, and high SEP?	Pupil premium funding was used as a school level indicator of SEP, as described in section 2.1, we have added to clarity to section 2.3 to make it clear how schools were categorised at low vs high as follows: “Taking this approach, we used pupil premium funding (see description in section 2.1) as a school-level indicator of SEP during stages 1 and 2, where the object of analysis is the school, not the individual. Schools were categories as low-SEP if the percentage of students eligible for Pupil Premium was below the county-specific mean and high-SEP if the percentage was above.” As stated in section 2.4.1 Demographic measures, individual-level SEP was calculated as follows:	284-288 319-324

		“Individual-level SEP was reported using the FAS, which is composed of six items relating to: (1) family car ownership, (2) holidays, (3) computers, (4) availability of bathrooms, (5) dishwasher ownership, and (6) having their own bedroom. These were used as a proxy measure of individual-level socioeconomic position by summing the answers (possible range 0–13), and dividing into predefined affluence groups (low = 0–6, middle = 7–9, high = 10–13).²³”	
24	How many schools were categorized as low versus high SEP?	Added to Table 2 in the results section. The recruitment process for GoActive was aimed at recruiting equal numbers of low and high SEP schools.	
25	What were the student characteristics of these schools overall (versus just the intervention participants)? This will help provide context for whether the participants were representative of the schools they were attending.	All students in Year 9 in GoActive intervention schools participated in the intervention, regardless of whether they participated in study evaluation measures. We have no comparable data available to compare the characteristics of students in the year group vs the evaluation participants.	
26	One area that is not fully addressed is intervention development, and the role this could play in intervention uptake and subsequent stages. It is unclear based from the Corder et al. paper whether students of lower SES were included in the GoActive intervention development process. This has major implications for whether the intervention would address stated interests/needs of students across the socioeconomic groups. Lack of representation of low SES groups in intervention development is an	We thank the reviewer for this comment and agree that lack of representation of low-SEP students is an opportunity for growth within school-based physical activity interventions. To address this comment we have added the following to the strengths and limitations section: “Data were not available to address other relevant questions, such as whether schools has access to facilities specifically needed to run GoActive (rather than general facilities) (to address stage 1), the SEP of schools that agreed to participate vs those who did not (to address Stage 2), the role the intervention development process could have played in the uptake of and engagement with the intervention (to address stage 5) or the cumulative effects of inequities across the stages of Love’s model.” And to the recommendations for future research and practice: “The lack of representation of low-SEP groups in intervention development is an opportunity for	722-728 782-784

	opportunity for growth within school-based PA interventions and contributes to the dearth of literature the authors state in the introduction. Although this is briefly addressed, I recommend the authors expand on this concept in the discussion/conclusion section.	growth within school-based interventions and an important area to be considered in the development and evaluation of interventions.”	
27	Be sure to define all acronyms throughout the manuscript. Throughout the manuscript, be sure there is consistency with the stated objective. For example, in the Abstract, the main objective of the paper is first stated as examining the “implementation and evaluation” of the GoActive intervention. However, the abstract methods then state that the intervention and research process were examined, and this seems to be more in line with what was done. The authors may consider calling this the “intervention process”, versus stating “implementation” for example. Implementation science is a process unto itself, and may be misleading to readers, particularly because there was actually little data presented about the implementation process (e.g., characteristics of the implementers, how well the intervention	We thank the reviewer for highlighting these. We have checked the acronyms are accurately defined throughout the manuscript. We have also update the abstract to reflect this comment, in addition to our stated aim at the end of the Introduction. We now consistently use the terminology “intervention and evaluation process” to avoid confusion with implementation science approaches.	

	components, etc. were implemented).		
28	Page 5, Line 11: The phrase “life chances” is unclear. Perhaps “all-cause mortality”? Page 5, Lines 31-34: I agree that there is a benefit to understanding the differential effects of PA interventions on youth from different socioeconomic groups. I think THIS is more the gap in the literature, rather than whether they would benefit from PA interventions (which is pretty well established). Page 6, Lines 7-9: Please clarify whether this refers to school-based PA interventions specifically. Page 6, Lines 9-16: Please elaborate on these socioeconomic inequalities – specifically, which socioeconomic groups it negatively affects. It can be assumed that those of low SES are negatively affected, but it would be helpful to clearly state this as it is the main premise of the study.	All remaining points addressed in the manuscript as follows: “Active adolescents experience better present and long term health and are more likely to become active and healthy adults.³⁻⁵” “However, across young people’s physical activity literature these studies have tended to focus on differential effects by gender.⁹” “Limited evidence from individual evaluations of school-based interventions and physical activity interventions document socioeconomic inequalities negatively impacting those of a low-SEP in the provision of, and access to, interventions and resources,^{16,17} variation in intervention uptake,¹⁸ differential intervention efficacy,^{17,19} differential long-term compliance²⁰ and differential response in evaluations.^{9,21,22}”	124-125 151-152 164-168
29	Please include the dates of the study.	Following added to methods section: “The GoActive trial was run between September 2016 and July 2018.”	211-212
30	Page 8, Lines 32-34: were the subgroup group analyses post hoc? Page 9, Lines 14-18: Please clarify whether this review of the	Clarification added as follows: “Subgroup analyses conducted as part of the trial evaluation reported a suggestion of a positive intervention effect among students of a low/middle-SEP.”	227-258

	literature and decision making was post hoc. Additionally, please expand on the “recent review evidence” and how that informed the comparison by low-SEP versus middle/high-SEP.	“In response to recent review evidence highlighted in the introduction⁶ (published after the main GoActive trial) and because of our focus on socioeconomic inequities we compare students of low-SEP to students of middle/high-SEP during stages 3, 4 and 5.” As highlighted in the introduction, “Recent review-level evidence highlights the importance of promoting and enabling physical activity among adolescents living in the context of socioeconomic deprivation, who report experiencing more barriers to physical activity when compared to other socioeconomic groups.⁶”	289-292 130-133
31	Page 10, Line 36: Please state in what population/setting this questionnaire has been previously used, as well as any other psychometric properties that are available. Was there an option for respondents to state, “I don’t know”? Page 10, Line 34: Please briefly describe the characteristics of contact teachers (e.g., role, years taught, age, etc.).	Reference to the SPEEDY study where the questionnaire was previously used has bow been included as follows: “These data were collected using a questionnaire previously used in the Year 9 data collection of the Sport Physical Activity and Eating Behaviour, Environmental Determinants in Young People (SPEEDY) study.²⁷” There was no option for respondents to state “I don’t know” Information added on the contact teachers as follows: “Data on the school physical activity policy and social and physical environment were self-reported at baseline by contact teachers (often Physical Education or Year 9 lead) at all schools.²¹”	339-341 333-335
32	Page 15, Lines 56-60: Examining differences by school-level SEP should be included in the analysis plan.	Described in section 2.9 as follows: “Data were explored using simple tabulations, graphical techniques and basic summary statistics and analysed using Kruskal-Wallis one-way analysis of variance by school-level SEP.”	436-438
33	Page 19, Line 34: Change “man” to “mean”	“man” changed to “mean” in text.	572
34	Page 23, Line 23: Change “wrist-worm” to “wrist-worn” Page 23, Line 27: Is “different” supposed to be “difference”?	“wrist-worm” changed to “wrist-worn” “different” changed to “difference”?	710 712

34	The first and third box appear to state the same thing. Please update if needed. Also, please be sure to define “NCD” within the figure legend.	Figure updated and definition added to figure legend.	
36	Tables and Figures: Please define SEP, SD, MVPA, and BMI	Definitions added	
37	Tables: Please take care to clarify whether “Low-SEP, Middle SEP, and High SEP” refers to individuals or schools, particularly with Table 3.	Clarification added in-text as follows: “Table 3 presents the uptake of the GoActive intervention by individual-level SEP using website engagement.”	516
38	Supplemental Table 1: please define acronyms and include n’s	Added to table legend	Supplementary File 1

VERSION 2 – REVIEW

REVIEWER	Johnson, Ashleigh San Diego State University, Exercise and Nutritional Sciences
REVIEW RETURNED	23-Feb-2023
GENERAL COMMENTS	Thank you for the opportunity to review a revised version of this manuscript. The authors have done a good job of addressing reviewer comments. I have only one minor comment: 1) Page 25, Line 693: This is the first and only time that COVID-19 is mentioned. I suggest removing it because it feels out of place.